# ProLo: Localization via Projection for Three-Dimensional Mobile Underwater Sensor Networks

**DOI:** 10.3390/s19061414

**Published:** 2019-03-22

**Authors:** Feng Zhou, Yushi Li, Hejun Wu, Zhimin Ding, Xiying Li

**Affiliations:** 1Acoustic Science and Technology Laboratory, Harbin Engineering University, Harbin 150001, China; 2Key Laboratory of Marine Information Acquisition and Security, Harbin Engineering University, Ministry of Industry and Information Technology, Harbin 150001, China; 3College of Underwater Acoustic Engineering, Harbin Engineering University, Harbin 150001, China; 4Key Laboratory of Machine Intelligence and Advanced Computing, Sun Yat-sen University, Ministry of Education, Guangzhou 510006, China; wuhejun@mail.sysu.edu.cn (H.W.); dingzhm3@mail2.sysu.edu.cn (Z.D.); 5Department of Computer Science, Sun Yat-sen University, Guangzhou 510006, China; 6School of Intelligent Systems Engineering, Sun Yat-sen University, Guangzhou 510006, China; stslxy@mail.sysu.edu.cn

**Keywords:** localization, underwater sensor networks, mobile node, global rigidity

## Abstract

We study the problem of three-dimensional localization of the underwater mobile sensor networks using only range measurements without GPS devices. This problem is challenging because sensor nodes often drift with unknown water currents. Consequently, the moving direction and speed of a sensor node cannot be predicted. Moreover, the motion devices of the sensor nodes are not accurate in underwater environments. Therefore, we propose an adaptive localization scheme, ProLo, taking these uncertainties into consideration. This scheme applies the rigidity theory and maintains a virtual rigid structure through projection. We have proved the correctness of this three-dimensional localization scheme and also validated it using simulation. The results demonstrate that ProLo is promising for real mobile underwater sensor networks with various noises and errors.

## 1. Introduction

The current advances in ocean sensor devices have enabled underwater wireless sensor networks (UWSNs) and mobile underwater wireless sensor networks (MUWSNs) to be designed for various underwater applications. For instance, sensor nodes can be deployed for the surveillance of harbors to avoid intrusions and monitor the pollution of ships. They have great potentials in many marine applications as they are significantly cheaper than using conventional oceanographic vessels. They have been proposed for fish farming, undersea earthquake forecasting, and tsunami warning [1,2,3,4].

Localization for the underwater nodes, especially mobile node localization, is one of the basic operations in the above applications [5]. Almost all underwater applications need to locate the targets and there is no Global Positioning System (GPS) available in underwater environments. In either MUWSNs or UWSNs, mobile node localization is needed. First, wireless sensor nodes are apt to drift with waves as they are usually not fixed by some wires, since they may be far away from shores. In contrast, boats can be tied to the berth using short ropes in a harbor. In addition, many applications need to monitor moving objects or require mobile underwater nodes to follow the objects or to patrol back and forth to cover a large area.

Although the autonomous underwater vehicles (AUV) are expensive, using MUWSNs would actually reduce the hardware cost due to the following reasons: (1) A mobile sensor node can cover a large space in the water, which otherwise should be covered by hundreds or thousands of stationary sensor nodes. (2) A mobile sensor node can go charge itself when its battery is low, whereas a stationery node cannot. Changing the batteries of a large scale stationery underwater sensor network is costly. (3) Mobile sensor nodes are able to drag a failure node back. This makes a team of mobile sensor nodes more robust to exceptions than a single node.

Therefore, in this paper, we mainly consider the problem of active mobile underwater node localization. The UWSN node localization problem is a subset of the MUWSN localization. Nevertheless, even the problem of UWSN localization is significantly difficult, as has been illustrated in previous studies [2,6,7]. There are much more issues to be explored in MUWSN localization. Although there are studies considering three-dimensional localization using projection [8], the problems of moving a whole underwater sensor network and the network structure maintenance among the sensor nodes have not been touched.

The more significant challenges in MUWSN localization are as follows: (1) The localization should consider the three-dimensional space movement instead of two-dimensional in traditional localizations. (2) The motion devices and sensors are inaccurate in measuring the direction and the speed of a sensor node and thus the positions cannot be calculated merely based on the measured information. (3) The acoustic transmission rate is relatively low. The centralized schemes and the schemes that require communication among different hops often cannot work. This is especially severe for the realtime moving nodes.

In this paper, we propose a range-based scheme of localization, ProLo. ProLo is capable of localizing nodes in a three-dimensional (3D) space using projection. Specifically, ProLo makes the three major contributions: (1) It requires only three beacons for underwater three-dimensional localization. (2)  ProLo is theoretically proved to ensure a node can be localized given the proposed minimal and necessary conditions in a 3D space. This is critical for the low data rate acoustic communications, as it significantly reduces the traffic compared to before. (3) ProLo is proposed with a concept of moving beacon plane that allows the sensor nodes with inaccurate motion measurements but still keeping the nodes localizable. Inaccurate motion measurements make it infeasible to localize a node using its measured moving distance and angles. The localization is performed by projecting the positions of the nodes onto the moving beacon plane and using the 2D rigidity theories for efficiency. The simulation results demonstrate that our algorithm could effectively handle the errors and localize the sensor nodes after they moved.

## 2. Preliminaries and Related Work

### 2.1. Problem

To focus on the localization, we omit the hardware differences among the acoustic sensors and make the following assumptions:Given a MUWSN, the sensor nodes communicate by direct single hop or indirect multi-hop acoustic signals. In this MUWSN, there are three beacon nodes whose positions are initially known during deployment. At least two of the beacons can communicate directly. Two nodes that can communicate directly are called neighbors in this paper. A beacon that is not a neighbor of other beacons can communicate with the other beacons through multi-hop message forwarding. The non-beacon nodes do not know their own positions but know the positions of the beacons through such message forwarding.A beacon has an acoustic communicating device underwater and GPS device above the water so that it can obtain the location and communicate with the other underwater sensor nodes.Within this MUWSN, two acoustic sensor nodes can measure the distances between them as long as they are direct communication neighbors. We omit the details of distance measurement techniques in this paper. The available distance measurement techniques can achieve an accuracy level of up to 96% [9].The velocities measured by the sensor nodes have errors. Nonetheless, the moving speed of the sensor nodes can be limited to a level that two sensor nodes moving towards the opposite directions can exchange at least a standard data packet.An underwater sensor nodes can measure its depth in terms of the distance from the water surface. On average, the normal depth sensors can achieve the accuracy level up to 99% [10].

Under the above assumptions, the MUWSN is deployed in a $E^{3}$ space (three-dimensional Euclidian space, also denoted as 3D); the entrance of this space is denoted as (0,0,0) and the exit point as (*x*, *y*, *z*). The sensor nodes should traverse from the entrance to the exit following the shortest path.

### 2.2. Rigidity

To solve this problem of localization for mobile nodes, we model a MUWSN as a dynamic graph that changes over time, since the shape of the MUWSN inevitably changes when the nodes move. At time $t_{0}$, the MUWSN network is modeled as a weighted graph, $G(t_{0})$. $G(t_{0})$ is a set of edges, vertices, and the edge weights. An edge weight is the distance between the two sensor nodes connected by the edge. The graph G(t) models the MUWSN of time *t*.

The nodes in a weighted graph can be localized if the graph satisfies certain graph rigidity conditions with a set of additional constraints. The reason is that the localizability of a node in a graph is closely related to the rigidity property of the graph, as having been proved in previous work [11,12,13,14]. Specifically, in $E^{2}$, the global rigidity can be used to determine the node localizability in a graph [15,16]. Therefore, in this paper, we still employ the concept of global rigidity to ensure the node localizability, but the problem involves a graph in the three-dimensional space instead of a graph on a two-dimensional plane.

The graph rigidity is usually defined using a *bar framework* concept. A bar framework *F*, *F* = (*G*, *p*), is a tuple that includes a finite graph *G* = (*V*, *E*) and a finite set of vectors *p* = ($p_{1}$, $p_{2}$, …, $p_{n}$) in $E^{d}$ where a vector $p_{i}$ corresponds to vertex *i* (i∈V and *n* = |*V*|) in *G*. The edges $e_{ij}$ ($e_{ij}\in E$ ) of *G* are called the bars connecting vertices *i* and *j* in *F*. *p* is named as the configuration of *F*. In framework *F*, The length of a bar $e_{ij}$ is also specified by the configuration *p*, as it can be calculated using the vectors $p_{i}$ and $p_{j}$.

Note that, under the above definition, a single graph *G* sometimes has different frameworks that share the same graph *G* but use different configurations. In effect, given the vertices, edges, and edge weights in a graph, the graph can be realized in different shapes in a space. A framework is such a shape realization of a graph.

A framework of a graph G(t) is denoted as F=(N,E,P), where N is the set of graph vertices and *E* is the edge set. An edge $e_{ij}$ denotes that nodes *i* and *j* (i,j∈N, e_ij∈E). P is called the configuration of framework F. $P=(P_{1},P_{2},\ldots ,P_{n})$ in $R^{d}$. Here, $P_{i}$ is the vector from the point of node *i* (i∈N) to the center of the shape of *F*, which is designated as **0**.

Two frameworks $F_{1}=\left(N,\;E,\;P\right)$ and $F_{2}=\left(N,\;E,\;Q\right)$ are equivalent if the follows are true: (1) $F_{1}$ and $F_{2}$ are formed by the same node set and edge set: N,E. (2) $\left|\right|p_{i}-p_{j}\left|\right|=\left|\right|q_{i}-q_{j}\left|\right|$, where i,j∈N and $e_{i,j}\in E$. Two equivalent frameworks $F_{1}=\left(N,\;E,\;P\right)$ and $F_{2}=\left(N,\;E,\;Q\right)$ are *congruent*, if their configurations P and Q are *congruent*. Two configurations P and Q are *congruent* if: |P|=|Q| and $\left|\right|p_{k}-p_{m}\left|\right|=\left|\right|q_{k}-qm\left|\right|$, k,m∈N and $e_{k,m}\notin E$.

A connected graph G(t) is rigid in $E^{d}$, if G(t) has a finite number of equivalent frameworks in $E^{d}$, where $E^{d}$ is the d-dimensional Euclidian space [17]. The graph shown in Figure 1a is a rigid graph in $E^{2}$ (two-dimensional plane), as it has finite number of frameworks (two possible frameworks in total) in a $E^{2}$ plane. In contrast, Figure 1b shows a flexible graph, as it has an infinite number of frameworks.

As can be seen in Figure 1a, the above generic rigidity is still not enough to localize a node in a network, since a rigid graph has several frameworks with different configurations. The configurations specify a number of candidate positions for a single node in the rigid graph, even the positions of other nodes are all fixed. In the graph in Figure 1a, moving the right corner node from the current position to the upper candidate position does not change the lengths of the existing edges of the graph. Consequently, it is possible to calculate two positions for the node. Its location cannot be determined. Next, we further confine the rigidity by global rigidity.

A graph instance G(t) of a MUWSN is globally rigid if: any two frameworks of G(t) are congruent. In a globally rigid graph, the configurations of all frameworks are the same. Hence, in the globally rigid graph G(t), each node has a unique coordinate candidate satisfying the distance constraints between this node and other nodes. The distance constraints are specified by the weights of G(t).

Our previous study [16] has shown that the nodes in a graph in $E^{2}$ are localizable, if the graph satisfies the following three conditions: (1) It is globally rigid. (2) It contains at least three beacons that are non-collinear. (3) Among the three beacons, at least two of the beacons are direct communication neighbors. The reason is that if any two beacons are not neighbors, the distance between the two beacons cannot be measured. According to the assumption, only direct communication neighbors can measure the distance between them. However, as shown in Figure 2, in a three-dimensional space $E^{3}$, the node localization conditions are more complex than in $E^{2}$.

It is difficult to determine whether a graph is globally rigid in the three-dimensional space. Furthermore, three non-collinear beacons in a globally rigid graph in $E^{3}$ are often not enough to determine the localizability of some nodes. To address these challenges, we use a moving beacon plane to project the MUWSN graph in $E^{3}$ to $E^{2}$. Subsequently, the moving control and localization can be performed virtually in a two-dimensional plane first. Then, the depth of each node is measured using its depth sensor.

### 2.3. Related Work

There have been great efforts devoted to the area of underwater localization [18,19]. The range-based methods use received signal strength or other distance sensors to measure the distances between nodes [20]. These studies introduce the helpful system developing experiences to our study. Among the various existing localization schemes, there are a few closely related studies in comparison with this paper.

Specifically, the following are two studies considering the node mobility. Yan et al. [21] proposed an asynchronous localization scheme that uses an iterative least squares estimator to predict the mobility of sensor nodes and localize the sensor nodes. Hu et al. [22] proposed a localization method using variational filtering. Their method utilizes the spatial correlation and temporal dependency information to improve the accuracy. The mobility pattern of the sensor nodes are modeled. With the mobility model, their method uses a variational filtering scheme to localize the nodes.

Several studies consider the fact that nodes can drift in real-world application scenarios. The drifting is caused by the interaction of different ocean currents and tides or internal waves [23,24,25]. Xia et al. [6] proposed a skeleton scheme that adopted the global rigidity to determine the set of localizable sensor nodes. With this localizable node set, an Analytic Hierarchy Process (AHP) evaluates the localization confidence of the sensor nodes.

Erol et al. [26] proposed a hybrid scheme to use Dive-and-Rise (DNR) beacons for underwater localization. These beacons periodically refresh their 2D coordinate vectors using GPS when they are on the water surface. Cheng et al. [27] proposed a quadrilateral localization scheme called underwater positioning scheme (UPS). It uses four beacons to localize an underwater sensor node. This UPS is suitable for stationery networks in 2D static environments.

Luo et al. proposed a localization scheme for double-head maritime sensor networks (LDSN) [5]. Their scheme contains three steps: self-moored node localization (SML), underwater sensor localization (USD), and floating-node localization algorithm (FLA). These three steps together perform the localization process so as to leverage the free-drifting movement of the sensor nodes. Zhou et al. [28] proposed a two-step hierarchical localization approach for the large scale UWSNs. This two-step approach refers to the anchor node localization and the ordinary node localization steps.

In comparison with the previous studies, this paper focuses on the following issues that have not yet been addressed well: (1) The mobility control of MUWSNs should be designed for node localizability and localization in the 3D underwater environments with the extremely low communication rate. (2) The realistic issues such as errors in moving speed and direction of the sensor nodes should be considered. (3) The localization scheme should be lightweight due to the limited communication bandwidth in underwater environments and the dynamics in moving. In the following sections, we present the details of our ProLo with the above underwater constraints.

## 3. Mobile Node Localization of ProLo

According to the assumption, there are at least three beacons initially deployed to form a triangle, since they are required not on the same line. As shown in the example of Figure 3a, the initial positions of the beacons are given. The non-beacon sensor nodes and these beacons, together with the wireless links between them constitute a three-dimensional fully connected graph.

Unfortunately, it is an NP-hard problem to determine whether the graph of a 3D MUWSN is globally rigid. To reduce the problem complexity, the dimension may be compressed. We first construct a virtual plane via three beacons that are non-collinear. This virtual plane is named as *beacon plane*, since the three beacons are on this plane. Recall that, in the problem assumption, sensor nodes can measure their depth. This ability can be used to measure the relative height of each sensor to the beacon plane.

It is feasible to project the coordinate vectors of the underwater sensors onto the beacon plane and obtain a new two-dimensional graph. The global rigidity of the projected graph on the two-dimensional plane can be constructed using our previous triangle extension approach (TE) [16]. The simplified process of constructing a globally rigid graph for a MUWSN is presented as follows.

In TE, initially two of the three beacons start a triangle extension operation. The two beacons are neighbors. They can use a few hops to communicate. Then, the neighbors of the two beacons perform the triangle extension operation. As shown in Figure 4, the extension includes four steps: (1) Node *a* extends r1 and r2 (r1 and B2 become *a*’s parents). (2) Node *b* extends r1 and r2 (r1 and B2 become *b*’s parents). (3) Node *c* extends *a* and *b* (*a* and *b* become *c*’s parents); (4) Finally, *d* extends *b* and *c* (*b* and *c* become *c*’s parents). With this series of TE extensions, the constructed graph can be proved to be globally rigid using the theorem of TE. The resulted graph from these steps is illustrated in Figure 4.

Upon TE is completed on the beacon plane, the projections of some underwater sensor nodes still might not be included in the globally rigid graph. These nodes are called the non-extension nodes. The reason for the existence of non-extension nodes is that these underwater sensor nodes may be deployed too far away to have enough neighbors to perform extension. This can be conquered by our designed mechanism: radial movement. First, the non-extension nodes move towards the center of the MUWSN until they find at least two neighbors whose projections belong to the global rigid graph. This is based on the assumption that the beacon positions are known to all of the sensor nodes in the MUWSN. As such, a non-extension node can extend two neighbors as its parent. Subsequently, its projection can join the globally rigid graph.

We call the projected globally rigid graph on the beacon plan as ProGR. In this ProGR, it is possible to calculate the projected coordinate vectors on the graph, since the candidate position for each point is unique according to the theorem of TE. Once the two-dimensional positions of each coordinate vector projection on the graph can be determined, the underwater sensor nodes can recover their three-dimensional coordinates: The three-dimensional coordinates of a sensor node can be composed from the two-dimensional coordinates and the height, since the height of the beacon plane can be obtained from the three-dimensional coordinates of the beacons.

The process of obtaining the ProGR in a MUWSN is as follows: First, the beacons should check whether they are on the same line. If they are on the same line, they should adjust their distances to form a triangle. The new positions of the beacons should be broadcasted to all of the nodes. Then, the center of the triangle named as beacon center is calculated and forwarded to other underwater sensor nodes. The hop of the beacons are set as 0. A non-beacon sensor node is one hop larger than the minimum hop count of its neighbors. The MUWSN is finally constructed when all of the nodes get the hop count or timeout occurs.

After the network construction, the nodes in the MUWSN starts radial movement if necessary. The radial movement is launched hop by hop, from the outermost hop to the hop nearest to the beacon center. On each node, the radial movement is directed by the distances between it and the neighbors with a smaller hop number. When this distance exceeds the predefined distance threshold, the node starts to move towards the neighbor until the distance is short enough that the node has a sufficient number of neighbors. A node that is not connected to any node in the MUWSN can also use radial movement as an attempt to find the network to avoid being lost.

The nodes then construct a 2D globally rigid graph on the beacon plane in a distributed manner using only the information of neighbors. Although it also uses TE, this globally rigid graph construction process is different from that in our previous scheme [16], in that an MUWSN is three-dimensional. To construct a globally rigid graph, the sensor nodes should first calculate the projections of the distances between themselves. The projected distance *v* is computed using Equation (1), where *d* is the distance and *h* is the difference between the depth of the node and the depth of the beacon plane. Note that the projection only uses the distance, as the position of the nodes are not calculated yet during the global rigid graph construction.

(1)$$v=\sqrt{d^{2}-h^{2}}$$

With the projected distances between the underwater sensor nodes, the sensor nodes are now able to construct the globally rigid graph. As proved in TE, the nodes in the globally rigid graph can be localized in a two-dimensional space. Using equations described below, the positions of the underwater sensor nodes can be calculated in the globally rigid graph in $E^{2}$. Then, the calculated 2D coordinate vector of a node can be upgraded to 3D via appending the depth of the node to the coordinate vector.

Then, the next step problem is to accurately and quickly calculate the 2D coordinates of the nodes in the globally rigid graph. The proved localizability property only ensures that the position of a node can be localized, but it does not provide the position. As the gradient descent methods or the neural network methods are time consuming (usually a few minutes), they are not suitable for the dynamic MUWSNs. The locations of the nodes in a MUWSN keep changing, due to the drifts or turbulence. Furthermore, as the nodes in a MUWSN move and stop periodically, the quick estimation of the positions leads to a shorter stopping period.

ProLo uses the direct computation using the geometry constraints on the beacon plane. Figure 5 shows the general scenario of the calculation for position candidates. In Figure 5, $m_{1}$ and $m_{2}$ are two position-known neighbors of node *q*. Their coordinate vectors are $\left\{x_{m_{1}},y_{m_{1}}\right\}$ and $\left\{x_{m_{2}},y_{m_{2}}\right\}$. $d_{1}$ and $d_{2}$ are measured distances between *q* and them. The equations of the two circles with radius of $d_{1}$ and $d_{2}$ in Figure 5 are listed in Equation (5). The line equation of $m_{1}m_{2}$ is:$$\frac{y-y_{m_{1}}}{x-x_{m_{1}}}=\frac{y_{m_{2}}-y_{m_{1}}}{x_{m_{2}}-x_{m_{1}}}$$

In Figure 5, the two lines $q^{\prime }q$ and $m_{1}m_{2}$ are perpendicular to each other, as $\Delta m_{1}qt$ and $\Delta m_{1}q^{\prime }t$ are congruent. This congruence can be simply proved by the congruence of $\Delta m_{1}qm_{2}$ and $\Delta m_{1}q^{\prime }m_{2}$. Consequently, $d_{m_{1}t}$ and from *q*, $q^{\prime }$ to *t* satisfy the two equations in Equation (2). Subtracting the first from the second equation in Equation (2), we get Equations (3) and (4).

(2)$$\left\{\begin{array}{c} d_{m_{1}t}^{2}+d_{qt}^{2}={d_{1}}^{2}\\ {\left(d_{0}-d_{m_{1}t}\right)}^{2}+d_{qt}^{2}={d_{2}}^{2} \end{array}\right.$$

(3)$$d_{0}^{2}-2d_{0}d_{m_{1}t}={d_{2}}^{2}-{d_{1}}^{2}$$

(4)$$d_{m_{1}t}=\frac{d_{0}^{2}-{d_{2}}^{2}+{d_{1}}^{2}}{2d_{0}}$$

(5)$$\left\{\begin{array}{c} {\left(x_{q}-x_{m_{1}}\right)}^{2}+{\left(y_{q}-y_{m_{1}}\right)}^{2}={d_{1}}^{2}\\ {\left(x_{q}-x_{m_{2}}\right)}^{2}+{\left(y_{q}-y_{m_{2}}\right)}^{2}={d_{2}}^{2} \end{array}\right.$$

Then, α is calculated using $\alpha =arccos\frac{d_{m_{1}t}}{d_{1}}$. The angle from $m_{1}m_{2}$ to the x-axis, β, is calculated by $\beta =arccos\frac{|m_{2x}-m_{1x}|}{d}$. Finally, the coordinates of *q* are obtained by Equation (6). Similarly, the coordinates of $q^{\prime }$ can be computed by Equation (7).

(6)$$\begin{array}{c} x_{q}=x_{m_{1}}+d_{1}cos\left(\alpha +\beta \right)\\ y_{q}=y_{m_{1}}+d_{1}sin\left(\alpha +\beta \right) \end{array}$$

(7)$$\begin{array}{c} x_{q^{\prime }}=x_{m_{1}}+d_{1}cos\left(\alpha -\beta \right)\\ y_{q^{\prime }}=y_{m_{1}}-d_{1}sin\left(\alpha -\beta \right) \end{array}$$

The above equations assumed the scenario that the coordinates of $m_{1}$ are smaller than those of $m_{2}$, as shown in Figure 5. The other scenarios of different values of $m_{1}$ and $m_{2}$ are omitted here, as they are similar and different only in the signs of some angles and coordinates. When α=0, the two circles only have a single intersection point, i.e., $m_{1}$, *t*, $m_{2}$ and *q* are all on the same line with $m_{1}m_{2}$. The above equations are still applicable. In total, the conditions for applying the above equations are the three localizability conditions mentioned in Section 2.2.

With one more neighbor knowing its location, the position of node *q* can be determined, since this one more neighbor cross-validates the correct position. The cross-validation process is as follows: The candidate positions of a node are used to calculate the distance between this node and the new neighbor. The correct position will introduce the least extent contradiction. Here, a contradiction means the inconsistence between the measured distance to a neighbor and the distance calculated using the candidate position of this node and the position of the neighbor. Even considering the errors in measuring, the absolute inconsistence of distances should not exceed the measured distance itself. For instance, as shown in Figure 5, $m_{3}$ is a new neighbor with known position. It validates that position *q* is the correct position for this node *q*. In contrast, the calculated distance between the candidate position $q^{\prime }$ and node $m_{3}$ is much larger than the measured distance between $m_{3}$ and node *q*.

The above process for localization is formulated in Algorithm 1. Initially, before localization, the globally rigid graph of MUWSN projections on the beacon plane are constructed so that the above localization equations can be applied. The globally rigid graph will be maintained in the algorithm so that the MUWSN nodes can be localized continuously. In the first part, the nodes calculate the coordinates of each MUWSN node with the coordinates of beacons and the hop distances between the MUWSN node and the beacons. Lines 23–28 are to calculate the exact values of the correct position coordinates. As shown on Line 25, the algorithm used the equations such as Equations (6) and (7) to calculate the candidate positions, according to the coordinate values of the two neighbors, $m_{1}$ and $m_{2}$. On Line 27, the algorithm chooses the best candidate that incurs the fewest contradictions among neighbors.

In the second part of Algorithm 1, the nodes start to move and adjust their positions if their positions exceed the threshold. Exceeding the threshold may cause other nodes to fail in constructing the globally rigid graph. The setting of this threshold is discussed next.

**Algorithm 1** ProLoAlg distributed localization and control algorithm.
1:on node *i*:2:running ← true3:**type** MUWSN-Projection: {id, state, p{x,y}, pt1, pt2, neighbors }4:// projection on the beacon plane, p{x,y}: the coordinate vector5://a sensor has 4 states: localized, flexible, rigid, localizable6:**type** neighbors[]:{id, state, d, p{x,y} }7://d: distance between the neighbor and this node8:moveid=0 // moveid: the times of moving epochs9:**while** running **do**10: //first part: localization11: **if** this node is beacon **then**12:  state ← localized13:  set up the beacon plane with the other two beacons14: **else**15:  **while** not receiving the beacon plane from beacons **do**16:   wait17: **while** globally rigid graph is not constructed **do**18:  //Use TE to construct the globally rigid graph on the beacon plane19:  **run** TE and initialize $r_{i}.p$ by distance vectors.20:  **if** no enough parents **then**21:   start contracting22: $r_{i}\leftarrow sensor-Projection$ //on this node *i*23: **if**
$r_{i}$.state is localizable **then**24:  broadcast state and calculate the position25:  candidates+= position() // use equations like Equations (6) and (7) to calculate new candidate positions26:  **if**
|candidates|>2
**then**
27:   ri.p← vote(candidates) //choose one candidate as the correct position28:   state ← localized29: // second part: moving control30: **if** MUWSN is localizing **then**31:  wait32: beacon starts moving: moveid ← moveid+133: **if** received moveid from neighbors and $p_{i}.moveid<moveid$
**then**34:  $p_{i}$.move($v_{i}$)35:  $p_{i}.moveid=moveid$
36:  state ← flexible37: **if** neighbor distance exceeds the threshold $T_{d}$
**then**38:  start contracting39: **if** state == flexible **then**40:  continue


In Algorithm 1, the threshold of the distance between two neighboring nodes is designed to keep the connection between two nodes while avoiding their moving collisions. Equation (8) shows the definition of this threshold. In this equation, $d_{m}$ can be either the minimum or maximum distance allowed, depending on which one leads to a smaller $T_{d}$. Choosing a smaller value as the threshold is to avoid either disconnection or collision. The smaller value obtained from the minimum allowed distance indicates that the distance is so short that the two nodes are about to collide. On the other hand, the smaller value obtained from using the maximum allowed distance means that the distance of the two nodes might exceed the communication range.

(8)$$T_{d}=\frac{1}{2}\min_{i,j\in N}abs\left(d_{ij}-d_{m}\right)$$

Suppose that the destination point of a node is possible to exceed the circle with the diameter of $T_{d}$ (Equation (8)), the node should start the position adjustment procedure. We use possible because an MUWSN node cannot move accurately in either direction or speed. The time of $T_{d}$ being exceeded can be predicted with this threshold. Usually, we define an epoch for each node to move within the period of the epoch. Altogether, considering the moving errors and the epoch length, Algorithm 1 is able to predicate whether the destinations of two nodes may cause the distance to exceed the threshold within the next epoch.

## 4. ProLo Algorithm Analysis

Three key implementation issues about ProLo must be analyzed to further investigate its realistic features. The first is the algorithm complexity. As the two algorithms LP and FC do not run in parallel, we analyze their complexity separately. The construction of the globally rigid graph in localization is a one-time operation performed only at initialization. The complexity of constructing the globally rigid graph is O(n) using TE [16]. The complexity of the localization part is $O(m^{2})$, where *m* is the number of direct communication neighbors.

The number of neighbors on average is approximately five. In total, the time complexity of localization makes it applicable to real-world scenarios. The motion control part performs the node control to maintain the global rigidity dynamically. At the start of each epoch, the motion control part controls the beacons and other nodes to move in turn. The time complexity of the moving control operation is O(1). At the end of each epoch (Δt), the motion control part calls the localization part to perform the localization.

The second issue to analyze is about the failures in a mobile MUWSN. The failures include localization failures and node failures. In some scenarios, the initial deployed network cannot be localized. For example, a node in a certain globally rigid graph may be assigned two or more different coordinates. Universal rigidity is not realistic for distributed algorithms: universal rigidity requires an excessively dense network and too much information among different hops.

This localization failure problem can be solved by the position adjustment operations. Assuming that the coordinates are correct, the localization failure node can move a few steps towards the virtual center. The node can finally determine which coordinates are correct, as moving towards the correct direction positively validates the correct coordinates.

Node failure can be addressed similarly via position adjustment. When a node failure breaks the global rigidity, the neighbors of the node start a *contracting* adjustment. From the border line of the network, the nodes move one step towards the virtual center, hop by hop, until global rigidity is recovered.

The third issue is the distance measuring error. The calculated position is based on the measured distance. Hence, distance error may cause localization errors. The localization error, in turn, may break the global rigidity when the nodes move. The distance measuring error is usually within 10% of the distance. To address this problem, the motion control part sets the epoch length to ensure that the distance measuring error is less than 10% of the motion error within an epoch. In addition, the distance measuring error is considered together with the motion error, and the epoch length is set within an appropriate range to avoid the motion control failure.

## 5. Evaluation

ProLo is designed for mobile node localization. Therefore, we tested both localization and moving of ProLo on a MUWSN. In the following, we first present the evaluation of ProLo on two different network configurations. The first was 50-node network including five beacons and the second was a 100-node network including nine beacons. We also compared ProLo with two classical localization algorithms, namely DV-distance and gradient descent localization, but the comparison was with the projected graph on the beacon plane in $E^{2}$. Their projection parts were carried out by ProLo.

Figure 6 shows the localization result of ProLo for a 50-node five-beacon three-dimensional MUWSN. As can be seen in Figure 6, the localization was performed through projecting the node-distances to the beacon plane. Then, the globally rigid graph was constructed on the beacon plane. It can be seen that all of the locations of the nodes could be calculated correctly. The sensor nodes communicated via multi-hops. The neighbors were connected by edges shown on the beacon plane.

Figure 7 and Figure 8 show the localization results of DV-distance and gradient descent algorithms for the projected 50-node five-beacon MUWSN on the beacon plane in $E^{2}$. Apparently, the accuracy of ProLo was the highest. The reason is that ProLo used geometry constraints to directly calculate the correct positions. The other methods using approximation could not get the same accuracy as ProLo.

Figure 9 shows the localization result of ProLo for a 100-node nine-beacon three-dimensional MUWSN. Most nodes could still be localized correctly. In comparison, the localization errors of DV-distance were much more severe, as shown in Figure 10. We were unable to get the results of gradient descent on this network configuration, as the algorithm did not converge due to the complexity of the neighbors in the network.

The above localization results demonstrate that the direct computation of ProLo was accurate, given the globally rigid graph and beacons. In contrast, the complex methods were inaccurate even though they are time consuming. Note that the simulation for stationery node localization did not include the distance measurement errors. The motion of the sensor nodes was not considered, either. They are presented in the following.

Table 1 lists the setup of the MUWSN being simulated in the moving evaluation. The network had 11 nodes to show the trajectories of the nodes clearly. The MUWSN nodes communicated using acoustic signals that can reach 1000 m. The area for the MUWSN to monitor was a square of 30 km × 30 km. The beacons were controlled to move from (0, 0, 0) to (30 km, 30 km, 0). The other nodes were controlled to follow the beacons while keeping their depth if possible. They could move upward or downward after initial deployment in the case there are obstacles.

The MUWSN could not be accurately controlled, due to the complex currents and drifts in oceans. To simulate the inaccuracy of movements, we added random errors to the motion sensors and distance measuring devices, as shown in Table 2. The minimum and maximum allowable distances between two nodes are also listed in this table. As the area was large and distances of the nodes may exceed the distance limits, multi-hop communications were needed from time to time.

We deployed the nodes in our own simulator. The nodes were controlled to maintain global rigidity while moving in simulation. The trajectory of the nodes from the entrance to the destination is shown in Figure 11. When large errors existed in the velocity and distance measurements, the nodes could not reach the destination accurately. Nonetheless, they could go somewhere around the destination. Moreover, Figure 11 shows that the nodes managed to adjust their positions to maintain the global rigidity of the network.

When to adjust the positions of the nodes to avoid disconnection or collision is an interesting topic in the evaluation of our localization-control algorithm. We performed a series of simulations at different moving speeds. The position-adjustment times are shown in Figure 12. The results indicate that the fastest speed caused frequent position adjustments. The optimal speed was approximately 1.7 m/s because the velocity error Was proportional to the speed. When a node was moving too fast, the node was likely to deviate from the intended trajectory, and the deviation might activate a position adjustment. In contrast, moving at a normal speed might slow the deviation. Moreover, the velocity error was random due to the complex environment; therefore, the errors at different periods might cancel each other.

## 6. Conclusions and Future Work

In this paper, we propose ProLo, a new distributed localization algorithm for MUWSNs. We define a beacon plane and project the edges of the non-beacon nodes to the beacon plane. This way, we reduce the problem from 3D to 2D. We apply global rigidity theory to enable the nodes to be localized during motion. The evaluation results show that our ProLo is capable of maintaining the global rigidity while the sensor nodes are moving in an MUWSN. The complexity and performances of moving and localization were analyzed. The position adjustment details were simulated and recorded.

ProLo cannot handle the distance measurement errors well, since it has to calculate the positions based on the measured distances. It is possible to perform cross-validation using probabilities of the distances between neighbors. We leave this as a future task.

## Figures and Tables

**Figure 1 sensors-19-01414-f001:**
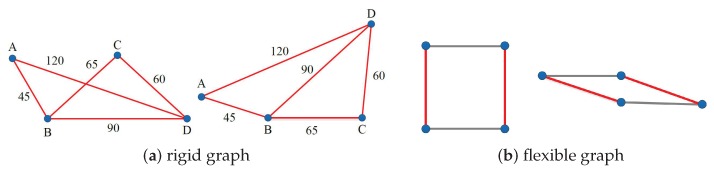
Frameworks of a rigid graph and a flexible graph.

**Figure 2 sensors-19-01414-f002:**
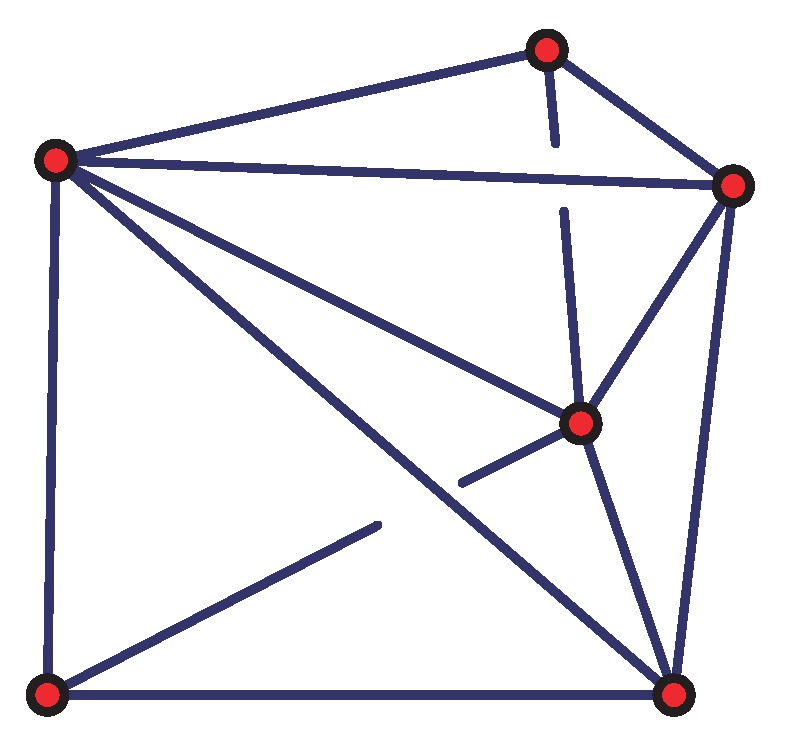
A globally rigid graph.

**Figure 3 sensors-19-01414-f003:**
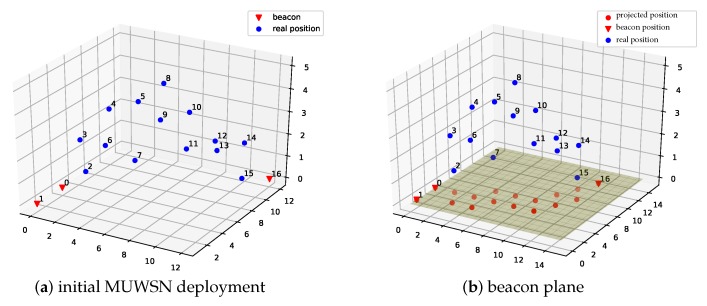
MUWSN network and beacon plane.

**Figure 4 sensors-19-01414-f004:**
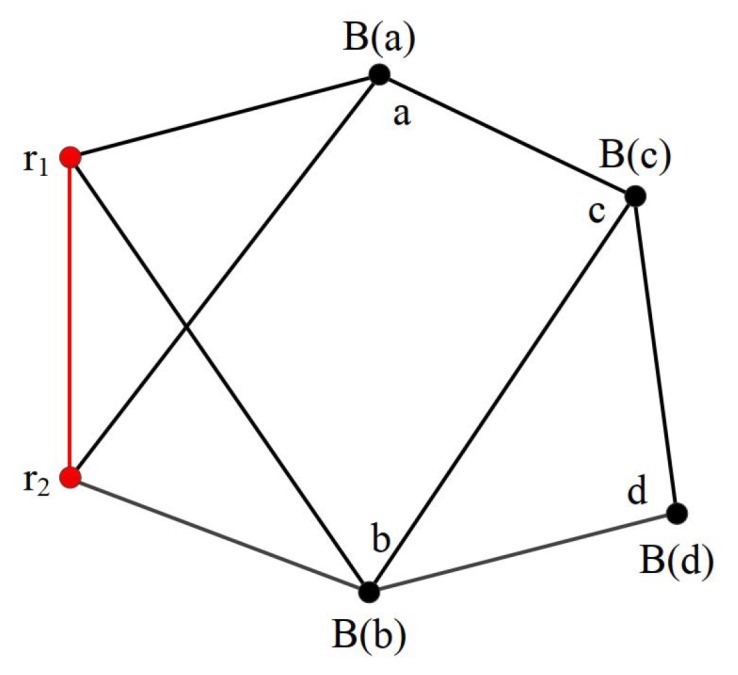
A globally rigid graph constructed step by step using TE.

**Figure 5 sensors-19-01414-f005:**
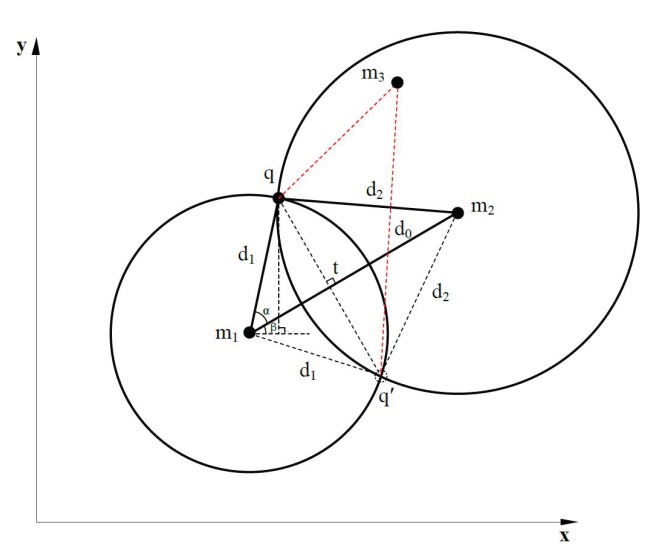
Position candidate calculation.

**Figure 6 sensors-19-01414-f006:**
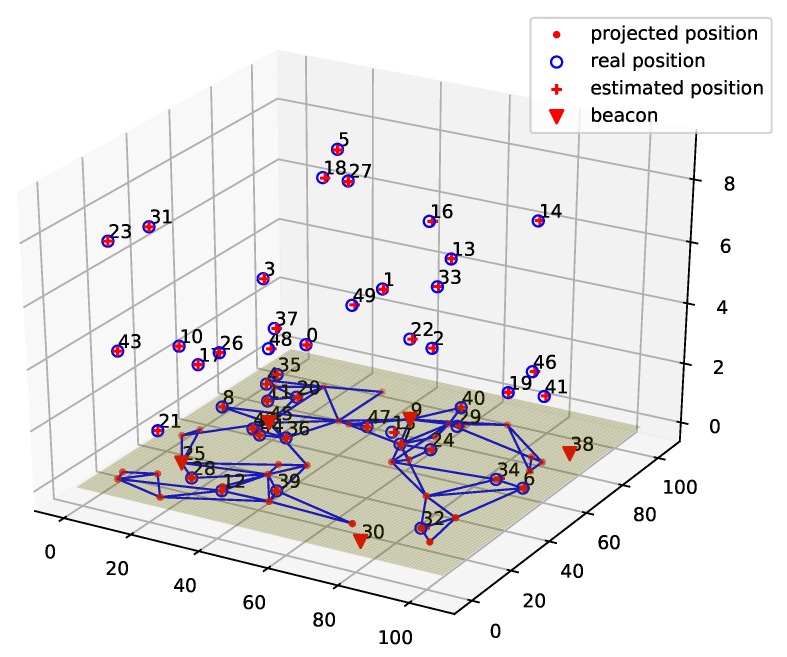
ProLo Localization for a 50-node 5-beacon 3D MUWSN.

**Figure 7 sensors-19-01414-f007:**
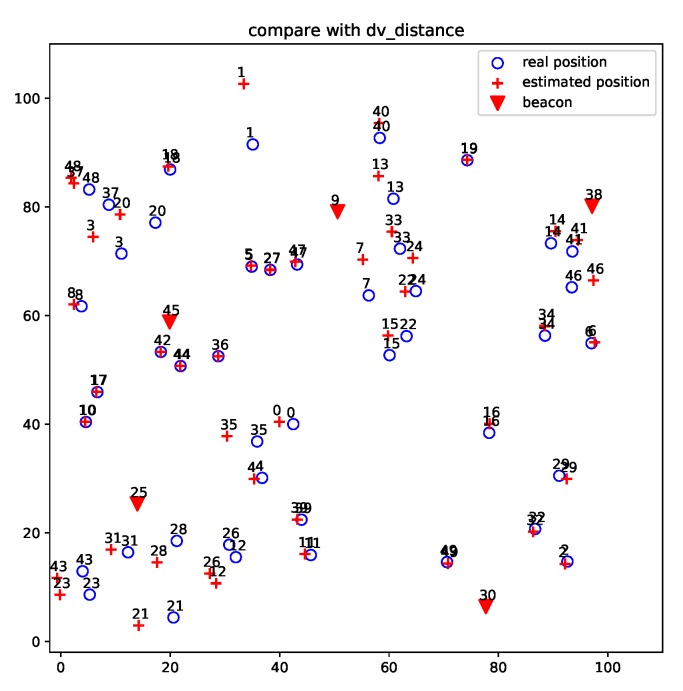
DV-distance localization for the projected 50-node 5-beacon MUWSN on the beacon plane in $E^{2}$.

**Figure 8 sensors-19-01414-f008:**
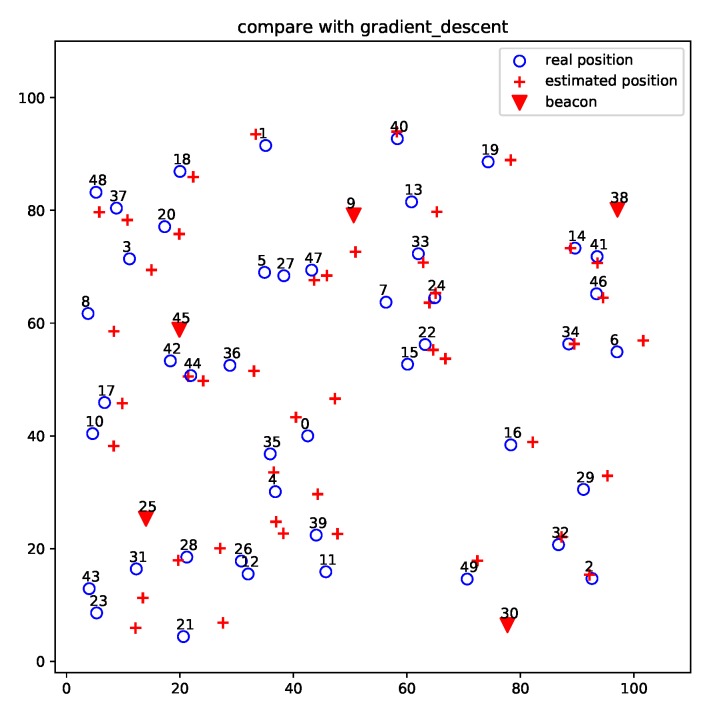
Gradient descent localization for the projected 50-node 5-beacon MUWSN on the beacon plane in $E^{2}$.

**Figure 9 sensors-19-01414-f009:**
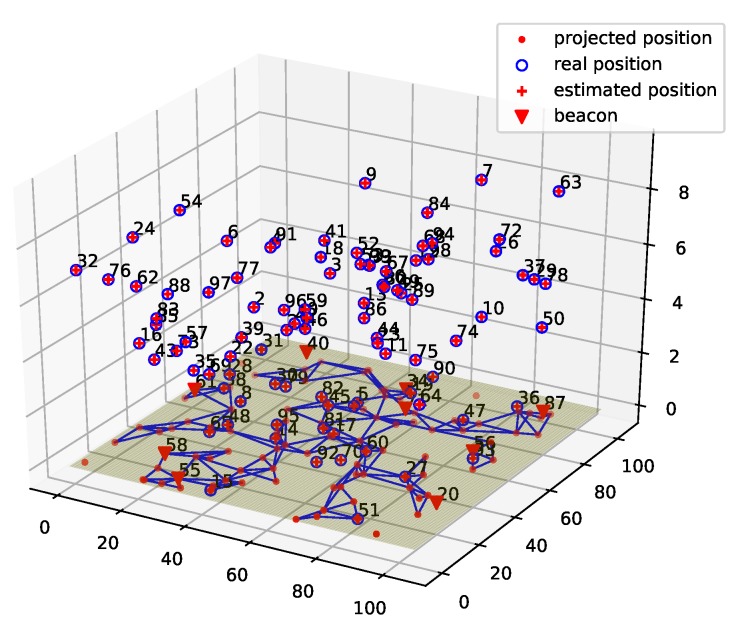
ProLo Localization for a 100-node 9-beacon 3D MUWSN.

**Figure 10 sensors-19-01414-f010:**
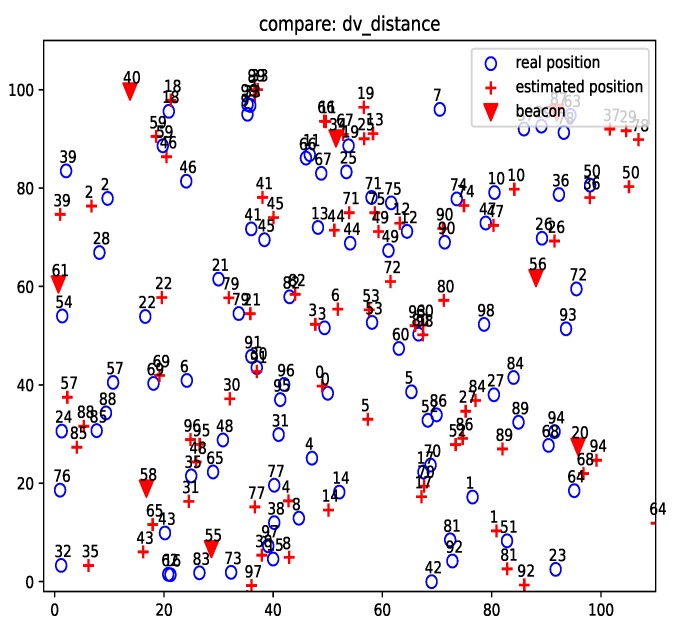
DV-distance result for the projected 100-node 9-beacon MUWSN on the beacon plane in $E^{2}$.

**Figure 11 sensors-19-01414-f011:**
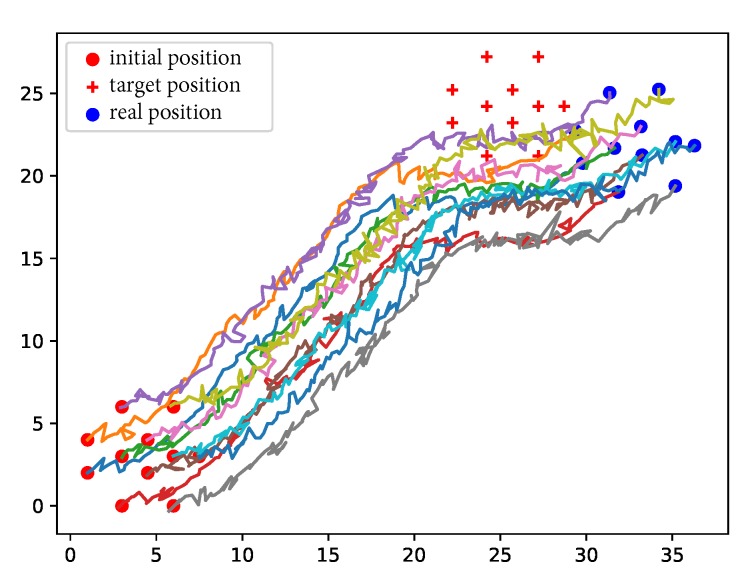
Moving node trajectory projections (km).

**Figure 12 sensors-19-01414-f012:**
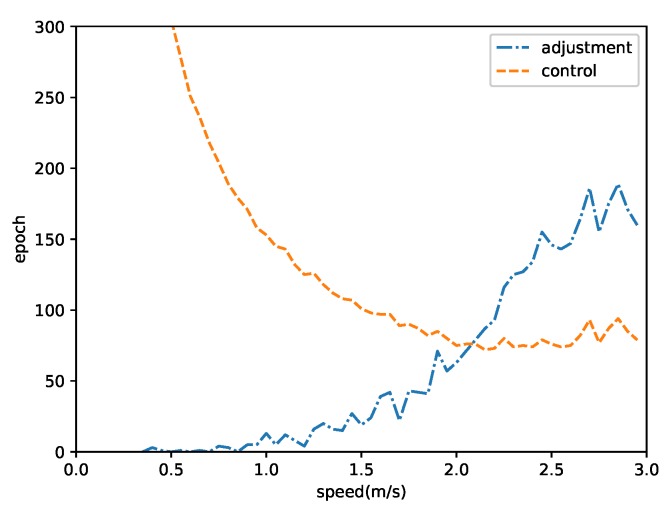
Position adjusting times.

**Table 1 sensors-19-01414-t001:** Experimental setup.

Num of Nodes	Num of Beacons	Area Width (km)	Communication Range (m)
11	3	30	1200

**Table 2 sensors-19-01414-t002:** Errors and distance limits.

Speed Error ϵ	Direction Error	Distance Error	Collision Distance (m)	Disconnection Distance (m)
±20%	±15	±5%	5	1000

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
