# Peer review of "ProLo: Localization via Projection for Three-Dimensional Mobile Underwater Sensor Networks"

_sensors, 2019, doi:10.3390/s19061414_

Reviewer 1 Report

This paper presents mobile UWSN localization algorithm in three-dimensional space. The preliminaries and related works are well written. The proposed algorithm (ProjGRL) is well  explained and organized. However, there is no comparison between proposed and conventional algorithms in the evaluation. The reviewer cannot judge superiority of the proposed algorithm. The other comments are follows:

(1) pp.6 3D localization in UWSN

The problem of 3D localization is also discussed in radio wireless sensor networks such as drones and unmanned aerial vehicles (UAVs). The reviewer thinks that the similar algorithms have been presented in those fields. Authors had better check these related works and assert which part is different in UWSN and other wireless sensors networks. The proposed algorithm seems to not be special to UWSN because underwater acoustic propagation is not considered.

(2) pp. 5 Evaluation

Authors mention “The other nodes are controlled to follow the beacons while their keeping depth if possible” in Line 287. The depths of UWSN nodes do not change in the simulation. It seems to evaluate 2D localization algorithm as the evaluation results are plotted in 2D plane. Authors had better evaluate the proposed algorithm in 3D space where x,y, z locations of UWSN nodes are changing. Authors had better introduce characteristic parameters in ocean environment. For example, random distributions of moving speeds are different in horizontal and vertical directions when considering actual ocean currents and drifts.

Author Response

This paper presents mobile UWSN localization algorithm in three-dimensional space. The preliminaries and related works are well written. The proposed algorithm (ProjGRL) is well explained and organized. However, there is no comparison between proposed and conventional algorithms in the evaluation. The reviewer cannot judge superiority of the proposed algorithm. The other comments are follows:

Answer: Thanks for the helpful comments. We are sorry that the original version did not present the comparison with the conventional algorithms. This time we added more evaluations in P.13-P.15.

(1) pp.6 3D localization in UWSN

The problem of 3D localization is also discussed in radio wireless sensor networks such as drones and unmanned aerial vehicles (UAVs). The reviewer thinks that the similar algorithms have been presented in those fields. Authors had better check these related works and assert which part is different in UWSN and other wireless sensors networks. The proposed algorithm seems to not be special to UWSN because underwater acoustic propagation is not considered.

Answer:  We are sorry that we did not make it clear in the original submission. There are indeed algorithms using projections in 3D. However, ours is different from them in that we constructed the rigid graph in the projection. This is new. The acoustic propagation is slow. Therefore, our localization uses the geometry constraints in the rigid graph to calculate the coordinates directly, as shown in P.9.

(2) pp. 5 Evaluation

Authors mention “The other nodes are controlled to follow the beacons while their keeping depth if possible” in Line 287. The depths of UWSN nodes do not change in the simulation. It seems to evaluate 2D localization algorithm as the evaluation results are plotted in 2D plane. Authors had better evaluate the proposed algorithm in 3D space where x,y, z locations of UWSN nodes are changing. Authors had better introduce characteristic parameters in ocean environment. For example, random distributions of moving speeds are different in horizontal and vertical directions when considering actual ocean currents and drifts.

Answer: Thanks for the insightful comments. The assumption is that the mobile nodes can measure their depths in the real node experiments. Actually, we bought an AUV and performed underwater test. It is possible to maintain a stable depth. This time we added random errors to the moving speeds and directions of the nodes. The result is shown in Fig. 12.

Reviewer 2 Report

Improve English. For example, 

line 24: "...vehicle... are expensive" instead of "...vehicles... are expensive"

line 26 " stationery" instead of "stationary"

line 33 "...studies consider..." instead of "...studies considering..."

etc.

Line 47: What does " sensor modes move inaccurately" mean? Inaccuracy can be applicable to measurement, not to motion.

Line 124. Should it not be: "(3) Each node has at least two neighbors..."?

Line 135: "previous"?

Line 144: "interaction"?

I don't see the difference between figures 3a and 3b.

There is no need to explain concepts like congruency and collinearity.

In Eqn.1, variables d and h are not explained.

In Figure 6, beacons' positions cannot be seen.

Figure 7 (and similar) have Cartesian coordinates, but scales for X and Y coordinates are different. This is confusing.

Figure 12: what is the difference between "object position" and "real position"? Is object not real?

I assume that "beacons" are on the surface and know their absolute positions using GPS. However, this is never stated explicitly.

Author Response

Thanks for your valuable comments, please see the attachment.

Round  2

Reviewer 1 Report

The reviewer confirmed that the paper is revised to reply the review comments and suggestions.

Author Response

We really appreciate your valuable comments .